# Assessment of the Relationship between Body Weight Status and Physical Literacy in 8 to 12 Year Old Pakistani School Children: The PAK-IPPL Cross-Sectional Study

**DOI:** 10.3390/children10020363

**Published:** 2023-02-11

**Authors:** Yinghai Liu, Syed Ghufran Hadier, Long Liu, Syed Muhammad Zeeshan Haider Hamdani, Syed Danish Hamdani, Shaista Shireen Danish, Syeda Urooj Fatima, Yanlan Guo

**Affiliations:** 1School of Physical Education, Shanxi University, Taiyuan 030006, China; 2Department of Sports Sciences, Bahauddin Zakariya University, Multan 60800, Punjab, Pakistan; 3School of Physical Education, Suzhou University, Suzhou 234000, China; 4Faculty of Sport Science, School of Kinesiology, Shanghai University of Sport, Yangpu District, Shanghai 200433, China; 5School of Sports Sciences, Beijing Sports University, Haidian District, Beijing 100084, China; 6Department of Physical Education, Government College University, Faisalabad 38000, Punjab, Pakistan

**Keywords:** physical literacy among Pakistani children, CAPL-2, correlation analysis, school children, childhood obesity, physical competence, daily behavior

## Abstract

(1) Background: Physical literacy (PL) is a multidimensional concept, since it fosters lifetime engagement in physical activities and reduces obesity; however, empirical evidence is lacking to support this association. This study first aimed to establish PL levels stratified by normal weight children and children with overweight and obesity. Furthermore, this study determined a correlation between PL domains and BMI by weight status among South Punjab school children. (2) Methods: This cross-sectional study involved 1360 (Boys: 675 and Girls: 685) children aged 8 to 12, and was conducted using CAPL-2. T-tests and chi-square were used to determine the difference between categorical variables, with MANOVA used to compare weight statuses. Spearman correlation was employed to determine the correlation between variables; *p* < 0.05 was considered significant. (3) Results: Normal weight children had significantly higher PL and domain scores, except for the knowledge domain. Most children with normal weights were at the achieving and excelling levels, while children with overweight and obesity were at the beginning and progressing levels. The correlation among PL domains in normal and overweight and obese children ranged from weak to strong (*r* = 0.001 to 0.737), and the knowledge domain was inversely correlated with the motivation domain (*r* = −0.023). PL and domain scores were inversely correlated to BMI, except for the knowledge domain. (4) Conclusions: Children with normal weight tend to have higher PL and domain scores, while those with overweight or obesity tend to have lower scores. There was a positive relationship between normal weight and higher PL and domain scores, and an inverse relationship was observed between BMI and higher PL scores.

## 1. Introduction

In the 21st century, the high prevalence of childhood obesity and overweight has become a global epidemic and a significant public health concern [1,2,3,4]. This problem is escalating globally and affecting both high- and low-income countries [5]. As the prevalence of obesity increases in low- and middle-income countries, obesity rates have been proven to be higher than in developed countries [6,7]. In 2016, the World Health Organization (WHO) reported that 20.8% of Pakistan’s population was overweight and 4.8% was obese, making it country severely affected by the obesity pandemic [8,9,10]. Similarly, Pakistan is equally affected by childhood obesity; the prevalence of overweight and obesity among children and adolescents aged 10 to 19 in Pakistan was estimated to be 10.7% and 6.6%, respectively, in 2018 [11]. There are many facets to the problem of childhood obesity; sedentary behavior and insufficient physical activity are major established contributors to childhood obesity [5,12,13], which causes long-term damage to a child’s health, including lower quality of life and adverse health outcomes in the future [14]. As a result, addressing physiological factors such as body composition is essential for effective obesity control strategies [15]; this highlights the pressing need for novel approaches to addressing children’s health problems.

According to a consensus-based definition, physical literacy (PL) is described as “the motivation, confidence, physical competence, knowledge and understanding to value and take responsibility for engagement in physical activities for life” [16]. PL supports lifelong physical activity (PA) and health by providing individuals with the knowledge, skills, and confidence to engage in a variety of PAs throughout their lifetime [17]; this includes developing movement skills, such as running, jumping, and throwing, as well as sport-specific skills, such as swimming or cycling [16]. Additionally, PL promotes the understanding of PA as a form of self-expression and encourages the integration of PA into daily life [18]. PL can lead to a greater likelihood of individuals engaging in regular PA, which can improve overall health and well-being.

In recent years, there has been a resurgence of interest in the concept of PL as a potentially helpful approach for understanding why certain children do or do not participate in regular physical activity [19]. According to the PL concept, acquiring fundamental movement skills is essential for developing an active and healthy lifestyle [20,21]. FMS are basic skills necessary for participation in PA and sport [22]. They include locomotor skills, such as running, hopping, and jumping; non-locomotor skills, such as balancing and stretching; and manipulative skills, such as throwing, catching, and kicking [23,24]. PL involves developing these FMS and the knowledge, attitudes, and behaviors that support their use [16]. Developing PL allows individuals to move confidently and competently in various PAs and environments, laying the foundation for lifelong PA and sport participation. Consequently, a physically literate child will be able to move confidently and efficiently in a wide range of physically demanding settings and respond efficiently to a wide range of physical situations [15,19]. Obesity has been recognized as a significant factor that negatively influences the individual PL journey, which can have far-reaching implications for the health of the individual [25]; hence, investigating how being overweight or obese affects children’s PL may encourage overweight or obese children to engage in a more physically active lifestyle [26].

The Canadian assessment of physical literacy (CAPL) was established in 2009 by the Healthy Active Living and Obesity research group (HALO) in collaboration with national and provincial organizations in Canada [27]. Instead of focusing solely on a person’s fitness levels, this multidimensional evaluation protocol provides a more comprehensive view of children’s health and active lifestyle behaviors [27,28]. The Canadian assessment of physical literacy-second edition (CAPL-2) aims to provide a valid, effective, and reliable assessment instrument for measuring children’s PL, aiding in the fight against and prevention of childhood obesity [27,28,29,30]. Only CAPL-2 encompasses the multidimensional aspect of PL, by merging multiple criteria into a unified, comprehensive evaluation [27,28]. CAPL-2 has four domains, included in accordance with the Canadian consensus definition of PL, the four domains are “Physical Competence, Daily Behaviour, Motivation and Confidence, and Knowledge and Understanding” [16]. Due to its strong connection with the concept of children’s PL, CAPL-2 has the potential to serve as a more comprehensive alternative to conventional physical fitness (PF) assessments [15]. For a more comprehensive overview of a child’s PL level, CAPL-2 provides composite scores and scores for each domain when assessing PL [31]. The CAPL-2 has many potential applications, including directing targeted wellness programs, providing insight into resource allocation, influencing policy decisions, and facilitating nationwide surveillance [27].

Previously, it has been suggested that PL may be associated with healthy weight status and various physical, behavioral, psychological, and social factors [27,32]. However, only two studies exploring this relationship have been conducted to date, both utilizing the CAPL-1 and CAPL-2 assessment tools. While the CAPL-2 is a newly developed tool, further international studies on different populations and cultures are needed to fully understand its utility in assessing PL and its relationship with weight status in various populations [29], particularly in Asian populations such as Pakistan, where no empirical evidence on this relationship currently exists. The current research gap in this area highlights the need for further investigation into the relationship between PL and weight status in Asian populations. Consequently, to accurately understand the relationships between PL domains in normal weight children and children with overweight and obese weight statuses, it is important to ensure that the CAPL-2 assessment tool accurately measures various PL components such as motor skills, physical activity behavioral, psychological, and social factors across all weight statuses [33]. Therefore, it is crucial to determine whether normal weight and overweight and obese children differ in their PL level and scores for each domain assessed by CAPL-2. Despite the importance of this topic, no previous studies have been conducted in Pakistan, to investigate the association between the PL domain scores and weight statuses as assessed by CAPL-2. Therefore, the first aim of this study was to examine the level of physical literacy and domain scores among 8–12-year-old normal and overweight and obese children from South Punjab, Pakistan, using the CAPL-2. The second aim was to investigate the relationship between PL and domain scores in children with normal weight and those with overweight and obesity. Lastly, this study aimed to determine the association between BMI, PL, and domain scores in this population.

## 2. Materials and Methods

### 2.1. Sample Size

This study employed stratified random sampling (SRS). South Punjab was selected as the population, and South Punjab’s three divisions (geographically subdivided by the government of Pakistan), namely Multan, Bahawalpur, and Dera Ghazi Khan, were considered as three stratums. Furthermore, 3 districts out of these divisions, Multan, Bahawalpur, and Dera Ghazi Khan, were chosen as strata, based on their population representation and geographic location within the province. The sample size of the current study was determined using Cochran’s (1977) formula [34,35]; previous studies accessing physical fitness characteristics among adolescents in South Punjab, Pakistan, also adopted a similar study design and sampling technique [36,37].
$$n=\frac{{\left(Z\right)}^{2}PQ}{e^{2}}\times D$$

Proportion (*P*) = 0.234

*Q* = 0.77 (1 − *P*)

*Z*-Score = 1.96 at a 5% level of significance.

*e* = 0.025 level of precision.

*D* = 5 (*D* is the design effect)
$$n=\frac{{\left(1.96\right)}^{2}\left(0.234\right)\left(0.77\;\right)}{{\left(0.05\right)}^{2}}\times 5=1359.8\;\approx \;1360$$

### 2.2. Participants

This study is a part of the Pakistan Initiative to Promote Physical Literacy (PAK-IPPL) study among children. This study employed a cross-sectional design to assess PL among 8-to-12-year-old higher secondary school children from South Punjab, Pakistan, using the CAPL-2. The research was carried out and finished during the academic year 2020–2021. In order to ensure representation of the overall population of South Punjab, 87 higher secondary schools were chosen from three strata: Multan, Bahawalpur, and Dera Ghazi Khan, with 29 schools in each stratum. However, two schools from the Bahawalpur and Dera Ghazi Khan strata refused to participate, resulting in a total of 85 schools being included in the study [38]. A sample size of 1360 students was randomly distributed across these 85 schools, using an equal allocation method. Initially, each school was asked to provide a list of students aged 8–12 years old, from which 16 students were selected randomly. However, during the field testing phase, 11% (150 participants) of the selected students either refused to participate or could not complete all the tests of the CAPL-2 protocol. To maintain the sample size of 16 students per school, these 150 participants were replaced with new samples taken from the same age and schools. The final sample of 1360 included 455 participants in Multan, 455 in Bahawalpur, and 450 in Dera Ghazi Khan.

Moreover, the children’s enthusiasm for learning and performing new tasks, as well as their desire to compete with their peers, played a significant role in motivating them to participate in the study. Additionally, the use of pedometers to assess daily activity levels increased the children’s interest in completing the task, which led to a high response rate for the completion of the study. Prior approval to conduct the study was obtained from the district’s education department. Due to cultural constraints, the researchers were prohibited from taking photographs or videos while collecting data. Additionally, to respect cultural norms, male study assistants were not allowed in female schools and vice versa during data collection.

Additionally, the consent of the parents of the children who volunteered was acquired in writing or verbally from each school’s principal. Data collection and testing were conducted during drill (physical education) classes, to avoid interference with the teaching schedule. Students were only permitted to participate in the study if they were able to carry out everyday activities normally, were psychologically healthy, had no evident physical defects, could correctly comprehend the test requirements, and were willing to comply with the completion of the tests.

### 2.3. Ethics Board Approval

The ethical board of the school of Physical Education of Shanxi University, China, approved this study in letters Dated: 2 January 2019 & 7 September 2022. Furthermore, to ensure that the study was conducted ethically and in compliance with local laws and regulations in Pakistan, approval was obtained from the ethics committee and office of the School Education department of South Punjab in letter#2189/GB.

### 2.4. Procedures and Measures

This study was conducted during the academic year 2020–2021. This study used questionnaires and other evaluation tools to collect information on demographic and anthropometric variables and to assess PL and domain levels. A schedule was followed during data collection; each school took 3 days to complete the assessment, and a schedule was followed during data collection to ensure consistency and efficiency in the data collection process. Specifically, each school was allocated 3 days to complete the assessment. This schedule was standardized across all locations visited by the researcher, as different cities were visited during the data collection process. To ensure the smooth execution of the data collection process, the data collection team contacted the decision-makers of schools in advance and inquired about the availability of student and guardian approvals. This technique helped to save time and ensure that all necessary approvals were obtained prior to the start of the assessment. On the first day, students were briefed about the purpose of the study and the CAPL-2 introduction, and the data collection team answered their queries. Then, a self-designed questionnaire and CAPL-2UQ (Canadian assessment of physical literacy 2nd edition-Urdu questionnaire) was delivered to the children, and a member of the research team assisted the participant if they had any trouble comprehending and completing the questions.

On the second day, the CAMS and a plank test were completed under the supervision of 5 appraisers. Before the assessment, an assessor demonstrated and instructed the students on how to perform the test, and each student was given one practice trial for the plank and two practice trials for the CAMS before the actual evaluation. On the third day, anthropometric data and the PACER test were completed, and the pedometer and its instructions and tracking log sheet were given to the participants. Missing pedometer day data were computed using the procedures described in the CAPL-2 manual [31]. If a participant could not attend school on the assessment day, we evaluated the missing test on the next day (since the data collection team visited each school at least four times) or the date for test evaluation was on the pedometer return day.

#### 2.4.1. Demographic Information

A self-developed questionnaire containing demographic information, such as name, age, gender, grade, and city, was distributed among children to collect participants’ demographic information.

#### 2.4.2. Anthropometrics

In order to accurately measure the children’s weight, height, and waist circumference (WC), children were instructed to remove their shoes and any other items that could affect the accuracy of the measurements.

Height: Height was measured using the metrical rod of a Digital Electronic Height-Weight Measurement Scale (Shanghai Puchun Measure Instrument Co., Ltd. Shanghai, China; Model NO.DT-150). The height measurement was recorded in centimeters (cm). During the test, the subjects were first asked to stand barefoot, with their back to the wall, standing on the flat machine surface, with a straight trunk and straight head.

Weight: Weight was measured using a Digital Electronic Height-Weight Measurement Scale (Shanghai Puchun Measure Instrument Co., Ltd. Shanghai, China; Model NO.DT-150). Weight was measured in kilograms (kg). Children were instructed to stand naturally in the center of the scale and maintain a solid body position. The examiner stood on the participant’s right side; when the decimal on the weight scale increased above 0.1 kg and the displayed value stabilized for five seconds, the examiner recorded the shown value [39].

BMI: The current study used the Centers for Disease Control and Prevention (CDC) standard formula and cut points (percentiles) to assess BMI and categorize the health status of participants [40]. BMI was calculated using the formula “weight (kg)/[height (m)^2^]” and is widely accepted as a reliable measure of health status [40]. According to the CDC percentiles, participants were classified as underweight if their BMI was <5th percentile, normal if it was between the 5th and 85th percentiles, overweight if it was ≥the 85th percentile, and obese if it was ≥the 95th percentile [40]. To classify children according to weight status, we divided them into two categories: normal weight children, and children with overweight and obesity. We then conducted analyses of the PL and domain scores within these two categories. The underweight children (*n* = 69) were removed due to their low percentage, as this practice was also used in a previous study [15,33].

Waist circumference: To measure WC, the examiner used a standard measuring tape and asked participants to stand straight, open their upper arms properly, close their feet, breathe normally, not purposefully contract or hold their breath, and distribute their weight evenly between both feet [41]. After exposing the abdominal skin, the examiner stood in front of the students and measured the horizontal position of the navel by one centimeter. The examiner’s eyes were on the same horizontal plane as the scale at the lower edge of the tape. The examiner paid attention to keeping the measuring tape parallel to the ground, recorded the reading, and kept the result to one decimal noted in centimeters.

#### 2.4.3. Physical Literacy

Children’s physical literacy was assessed according to the CAPL-2 protocol [31], which is reliable and has been validated in many countries [42,43,44]. This study assessed PL and domains using the CAPL-2 (Figure 1). Children were tested on their physical abilities, daily behaviors, motivation and confidence, and knowledge and understanding of PA and healthy lifestyle practices through these four multidimensional and overlapping concepts. According to HALO [31], the composite PL scores results from various domains, such that the daily behavior domain serves as the behavioral outcome of the other three domains. For example, a more significant number of daily steps may affect a child’s motivation and self-confidence to lead an active lifestyle, so a complete and proper assessment is necessary.

The total PL scores for each child were converted to a score out of 100 by allocating 30 points to the physical competence (PC) and daily behavior (DB) domains, 30 points to the motivation and confidence (M&C) domains, and 10 points to the knowledge and understanding (K&U) domains. Composite PL and domain scores were calculated following the above distribution, and CAPL-2 proposed an additional breakdown of the PL and domain scores into four categories, corresponding to the four stages of physical literacy development: beginning, progressing, achieving, and excelling. For instance, interpretation of these categories means that a child being at the beginning and progressing phases of PL indicates that they have not yet reached the recommended level of PL. At the same time, being in the achieving and excelling stages shows that they have achieved the recommended level. The procedures used in this study were in accordance with those outlined in the CAPL-2 manual and those used in earlier studies [31,45,46]. Total PL scores were also generated for each domain using the methods outlined in the test manual (see Figure 2). Domain scores could be calculated using the procedure outlined in the test manual, even if a single domain measurement was missing [31]. When the result of any measure is missing or withdrawn, an algorithm generates a total domain score that incorporates all measurements [31]. The PL domains and testing procedures employed in the current study are briefly outlined below.

#### The Daily Behavior (Behavioral Domain)

The daily behavior domain employed two methods to assess DB patterns regarding physical activity and had 30 points, with daily step counts accounting for 25 and self-reported PA accounting for the remaining 5 points. An objective method used pedometers to count steps, and a subjective method used a self-administered question on how many days a person engages in PA (MVPA) for 60 min per week. Daily steps were measured using Yamax DigiWalker SW-200 pedometers (Yamax Corporation, Tokyo, Japan). The participants were given pedometers to wear over their right side hip bone, fastened to the waistband of their trousers (pants for boys and shalwar for girls) for seven consecutive days. The first day the children received their pedometers was designated as a “practice day”. For the following seven days, except for the day used for practice, kids or parents recorded their daily step count before turning it in for the night. They recorded the entire time the pedometer was removed throughout the day on the log sheet. Moderate to vigorous physical activity (MVPA) was measured with one item from the CAPL-2 questionnaire. Participants self-reported the number of days they engaged in MVPA for at least 60 min per week [31,46].

#### Physical Competence Domain:

Physical competence was evaluated using three separate tests. The PC domains comprised a maximum of 30 scores (10 scores per test), and all tests were evaluated and scored from 1 to 10 points, based on the children’s performance according to the criteria outlined in the test manuals.

The PACER (20 m run) test measured cardiorespiratory competence (aerobic endurance) [47,48]. The test required children to run back and forth on an empty 20 m track with cones at each end. The child’s performance was measured by the number of laps he or she completed.The abdominal plank test measured muscular endurance (torso strength) [37,49]. The plank test measured muscular endurance by requiring children to maintain their body posture for as long as possible by stretching their legs and lifting their knees, adopting an upright position supported only by their forearms and toes, and adopting an upright position.The Canadian agility and movement skill assessment (CAMSA) test was used to assess agility and mobility skills [50]. A wide variety of fundamental motor abilities, such as throwing, kicking, skipping, catching, sliding, and hopping, were evaluated as part of the CAMSA [31,39]. There were fourteen distinct tasks that the contestant needed to complete perfectly in order to achieve maximum points (14 points total). Each skill was graded according to whether or not the participant matched the skill’s preset mobility criterion. They received a maximum score of 14 if no errors were made for any of the criteria. Before taking actual measurements, all physical tests were available for practice to all contestants. The best score of the two test trials was used to score the final result.

#### The Knowledge and Understanding Domain (Cognitive Domain):

The cognitive domain was assessed using CAPL-2 knowledge and understanding questions. This domain accessed children’s K&U of how to enhance physical competence, how to execute and improve daily PA routines, and issues such as what is muscular endurance and cardiovascular fitness, which were all addressed and evaluated in this questionnaire [40]. The K&U domain consisted of a total of 10 points, which comprised five questions, the first four questions of which were multiple-choice (1 point per question; a total of 4 points) and the fifth of which was a fill-in-the-blank format (1 point per fill-in-the-blank space; 6 points in total).

#### The Motivation and Confidence Domain (Affective Domain)

In the affective domain, four motivation and confidence aspects were measured: “predilection, adequacy, perceived competence, and intrinsic motivation” [31]. The M&C domain was assessed through 12 questionnaire items, each of which was given 2.5 points for a total of 30 points. Each effective domain aspect was measured with three questions, each worth 2.5 points, for a total of 7.5 points per M&C aspect [51,52]. Children’s self-predilection and adequacy assessed their physical ability and preference for an active lifestyle over a sedentary one and their potential to be successful. The section on self-perceived competence and intrinsic motivation asked participants about their confidence in their ability to achieve their goals and was evaluated using the concept of adequacy. On the other hand, intrinsic motivation measured the extent to which children are engaged in PA because they enjoy it, rather than because they believe it will help them avoid parental or teacher pressure.

### 2.5. Statistical Analysis

The present study used IBM SPSS version 22 (Armonk, NY, USA: IBM Corporation) for data analysis. Descriptive and inferential statistics were used to present the data, including means, standard deviations, and percentages. Outliers were identified using the standardized variable approach, where any observations deviating more than ±5 Z-scores from the mean for all measured variables were considered outliers. However, no specific outliers were identified; thus, the entire sample of 1360 children was used for data analysis. The normality of the data was determined through the utilization of Q–Q plots and histograms, which have been widely accepted in the literature as reliable methods for assessing normality [53]. An independent samples *t*-test was employed to investigate differences in gender across age, height, weight, and WC. The chi-squared test was employed for categorical variables to identify differences. A multivariate analysis of variance (MANOVA) was used to determine the differences among demographic characteristics, overall PL, and domain scores between normal weight and overweight and obese children. Spearman correlation was used to investigate the association between categorical variables (measured on an ordinal scale). Pearson’s correlation determined the relationship between BMI, overall PL scores, and domain scores. Multivariate logistic regression was used to determine the odds ratio (with a 95% confidence interval). In this study, Partial Eta Squared (η_p_^2^) was used to determine the effect size: a small effect ≤0.01, a medium effect ≤0.06, and a large effect ≥0.14 [54]. The significance level was considered as a *p*-value less than 0.05.

## 3. Results

Table 1 and Table 2 presents a comprehensive demographic analysis of the gender-specific anthropometric variables among the 8-to-12-year-old children who participated in this study. The mean age of the sample population was 10 years. The mean and SD values for both genders’ height, weight, BMI, and WC were 137.26 cm, 30.58 kg, 16.05 kg/m^2^, and 59.22 cm, respectively. Furthermore, the data revealed minimal variation in the average height of both genders. However, a significant difference emerged when analyzing other anthropometric variables, such as weight and WC. Specifically, boys exhibited higher mean values for weight and WC compared to girls. Conversely, girls exhibit a higher mean value for BMI compared to boys. A *t*-test analysis of these data demonstrated that there was a statistically significant difference between WC for boys and girls (*p* < 0.001), while there were no significant differences between the other anthropometric variables.

According to the data presented in Figure 3, the overall PL level of children with a healthy weight in South Punjab fell within the “progressing” and “achieving” categories, with 40.7% and 17.7% of participants, respectively. Conversely, overweight children tended to demonstrate lower PL levels, with the majority falling into the “beginning” (4.8%) or “progressing” (7.7%) categories across all the domains. Overall, the majority of children in both the normal weight and children with overweight and obesity were at the “progressing” level of PL, as determined by the CAPL-2 interpretation categories.

Table 3 presents the correlations between the PL domain scores for children who were classified as having a normal weight. The data reveals that there was a moderate correlation between physical competence and daily behavior (*r* = 0.583, *p* < 0.01), as well as between physical competence and motivation and confidence (*r* = 0.464, *p* < 0.01) in healthy-weight children. Additionally, a moderate correlation was found between motivation and confidence with physical competence (*r* = 0.448, *p* < 0.001). However, a weak correlation was observed between knowledge and understanding, daily behavior, physical competence, and motivation and confidence (*r* = 0.002, *r* = 0.001, and *r* = 0.006, respectively).

Table 4 presents the correlation coefficients between the scores of the PL domain in overweight and obese children. The results indicate that physical competence exhibited a strong positive correlation with daily behavior (*r* = 0.737, *p* < 0.01) and a moderate positive correlation with motivation and confidence (*r* = 0.496, *p* < 0.01). On the other hand, knowledge and understanding were weakly positively correlated with physical competence (*r* = 0.089) and moderately positively correlated with motivation and confidence (*r* = 0.104). However, it is worth noting that knowledge and understanding were negatively correlated with motivation and confidence (*r* = −0.023).

Table 5 presents the results of the Pearson’s correlation analyses, which revealed the association between BMI, total PL scores, and domain-specific PL scores. The correlation coefficients indicated that there was a statistically significant but weak inverse correlation between BMI and total PL scores, as well as domain-specific scores, with the exception of the knowledge and understanding domain, which demonstrated a very weak positive correlation with BMI (*r* values ranging from −0.171 to −0.217, with the exception of *r* = 0.006 for the knowledge and understanding domain).

Table 6 presents the gender-specific demographic information and descriptive analysis of anthropometric variables for children with normal weight and children with overweight and obesity. In the current study, most children in South Punjab had a normal weight (*n* = 1085). The distribution of boys and girls in the normal weight and overweight and obesity groups was found to be almost equivalent (i.e., *p* = 0.832). Furthermore, the age and height of children with normal and children with overweight and obesity statuses differed significantly. In the normal weight category, children had a lower weight (28.86), BMI (15.39), and WC (58.33) than the children with overweight and obesity, while the weight, BMI, and WC among normal weight children and children with overweight and obesity showed a significant difference (*p* < 0.001).

Table 7 presents the results of the total PL scores, as well as the scores of the four domains for children with healthy and overweight status. A Multivariate analysis of variance was conducted to determine the significance of the differences between the scores of these two groups. The results indicated that the differences between the scores of the healthy and overweight children were statistically significant for all domains except for the knowledge and understanding domain (*p* < 0.001). Furthermore, the mean scores for total PL and the other three domains (DB, PC, and M&C) were found to be higher among children with healthy weight, with the exception of the knowledge and understanding domain, where overweight children had higher scores. Overall, these findings suggest that children with healthy weight tend to have better PL scores and scores in the other three domains, except for the K&U domain. The differences between the scores of the healthy and overweight children were significant (*p* < 0.001).

## 4. Discussion

The first aim of this study was to determine the current status of physical literacy, and the second aim was to examine the correlation between PL and various domains in 8–12-year-old children in South Punjab, Pakistan, stratified by weight status (normal and Children with overweight & Obesity) and using the CAPL-2. Lastly, this study also examined the relationship between BMI, PL, and domain scores. The results regarding the PL status show that normal-weight children achieved a higher level than overweight and obese children in all domains, with a significant difference found between the two groups, except in the knowledge and understanding domain.

The current study established the level of PL for 8–12-year-old Pakistani children, and the results showed that 31.3% of normal and 3.5% of children with overweight and obesity were found to be at the achieving and excelling levels of PL (the highest levels of PL [31]). In comparison, 52.7% of normal children and 12.5% of children with overweight and obesity were at the beginning and progressing levels of PL (the lowest levels of PL [31]). While comparing the results of our study with other studies, similar findings were found in a Canadian study [33], where children with normal weight reported higher PL and domain scores than children with obesity. Another recent study on Spanish children found that 63.9% of non-overweight children attained the achieving and excelling level of PL [15]. A few other studies that did not stratify participants by weight status reported that most participants achieved beginning and progressing levels of PL [43,44,55]. Overall, our study results indicated that a higher proportion of Pakistani children with normal weight achieved an acceptable level of CAPL-2.

This study found a moderate to strong correlation between physical competence and daily behavior (*r* = 0.583 and 0.737, respectively), motivation and confidence, and daily behavior (*r* = 0.464 and 0.496, respectively). In addition, we found similar moderate correlations (*r* = 0.448 and 0.526, respectively) between motivation and confidence and physical competence among children with normal weight and overweight and obesity. Additionally, a weak inverse correlation (*r* = 0.006 and −0.023) was only found among the knowledge and understanding and the motivation and confidence domains among the normal and children with overweight and obesity. The results indicate that motivation and confidence are significant correlates of daily behavior and physical competence [15,33]. These findings are promising, since they show the importance and the necessity of developing M&C-focused interventions for combating and controlling obesity in children and adolescents.

This study found a moderate correlation between the physical competence and daily behavior domains in children with a normal weight and a strong correlation in children with overweight and obesity. This suggests that there is a link between a child’s physical competence and their daily behaviors. This relationship was statistically significant for both groups and is supported by previous studies [56,57,58]. For instance, a study conducted in Canada identified an association between the domains of DB and PF in children who were either of normal weight or overweight [33]. Additionally, other studies that were not conducted using the CAPL assessment tool also yielded similar findings. Another study, which was cross-sectional in nature, revealed a significant relationship between DB and aerobic fitness in children from England [55]. The study also suggested that a less active daily lifestyle is linked to lower physical fitness levels and an increased risk of chronic diseases in adulthood. In addition, a study conducted by Mak et al. studied the relationship between health-related physical fitness and weight status among Hong Kong children and adolescents aged 12 to 18, and found a significant correlation and impact of weight on PF [59]. This suggests that weight status can significantly affect an individual’s PF. Moreover, a systematic review reported that there is a positive relationship between daily behavior (accessed by PA) and physical competence (measured by FMS). [60]. This highlights the importance of regular physical activity in maintaining physical competence.

Similarly, our study found that there was a moderate correlation between motivation and confidence and daily behavior for both normal and children with overweight and obesity. The correlation was similar for both groups, and there were no significant differences between them. These results are consistent with the findings of a Canadian study which also observed a moderate association between PA and psychological factors (similar to M&C constructs) among normal weight and unhealthy-weight children [33]. Furthermore, our findings are consistent with the existing non-CAPL literature, which reports a low level of association between M&C and DB domains. For instance, a study conducted among 11-to-19-year-old Belgian school children assessed differences in PA habits, sedentary behavior (SB), and psychosocial variables (such as motivation, self-efficacy, and confidence level) among healthy and unhealthy children. The results revealed that the healthy weight children exhibited higher levels of these psychosocial variables compared to the unhealthy-weight children. Similarly, the study found a moderate correlation (*r* = 0.20 to 0.41) between PA and psychosocial variables in both weight groups, similarly to the findings of our study [61]. A study conducted in Canada found a moderate association between the M&C and DB domains of CAPL-1 among healthy and overweight/obese children, with correlation values ranging between 0.35 and 0.34 [33].

The correlation between the motivation and confidence and the physical competence domains was found to be moderate and similar among children with normal weight and those who were overweight or obese. Despite the limited research on this type of association, the findings were consistent, as demonstrated by Delisle et al., who reported a moderate correlation (*r* = 0.42–0.39) between M&C and PC using the CAPL-1 assessment tool, with no significant differences observed between the two weight groups [33]. A few other studies accessing the association between M&C and PC, and utilizing the same variables with the CPLA-2, have shown similar moderate correlation results (for plank, *r* = 0.23; CAMSA, *r* = 0.31) [29,49,50].

A weak correlation was observed between knowledge and understanding and the other domains of PL in normal weight and overweight/obese children [15], except for the overweight/obese children, where a weak inverse correlation was observed between M&C and K&U. There are several possible justifications for the observed weak correlation between K&U and other domains of PL. One possible explanation is that PL is a multidimensional construct that encompasses not only K&U but also PC, DB, and M&C. Thus, even if an individual has a high level of K&U about PA and PC, they may still lack the PC, positive attitudes, and M&C necessary to be physically active. However, the reason for this inverse correlation among the Pakistani population requires further investigation. Additionally, another possible explanation is that K&U alone is insufficient to promote physical literacy, and that other factors, such as social and environmental factors, also play a role. An American adolescent study assessed the impact of knowledge on physical activity (PA) lifestyle and its role in effecting change. The findings indicated that knowledge alone was insufficient for altering PA behavior [62]. Thus speculatively, it is possible that low motivation among overweight/obese children stopped them from making positive changes, despite having higher knowledge scores.

Furthermore, it can be suggested that better scores in the motivation and knowledge domains could lead to a higher level of PL in children. Thus, the positive correlations between the M&C and K&U domains led to a higher level of PL in normal weight children than in children with overweight and obesity [33]. Given the lack of existing research on this domain, it is recommended that future investigations be conducted to further explore and expand upon the current understanding of this relationship.

The results indicated a weak to moderate inverse correlation between BMI and levels of PL and its domains, with the exception of the knowledge and understanding domain. BMI was found to be negatively associated with higher scores for PL and its domains, consistently with previous studies that reported similar negative associations and concluded that normal weight children tend to achieve higher scores in PL and its domains [15,33,63]. Additionally, a subsequent study utilizing PLAY tools revealed a negative correlation between body composition and health-related quality of life, consistently with the findings of our study [63]. The findings of the current study in the domain of K&U revealed a weak correlation (*r* = 0.006). While knowledge is widely acknowledged as a crucial determinant of engaging in an active lifestyle, previous research also highlighted the significance of other factors, such as motivation and confidence, in shaping physical activity behavior [15,56]. Thus, it is clear that knowledge alone is not sufficient for promoting an active lifestyle.

The results of the daily behavior domain indicate that normal weight children obtained higher scores (11.06), and a significant proportion of normal weight children (30.5%) attained achieving and excelling levels compared to children with overweight and obesity. This finding is consistent with previous studies. For example, a literature review reported that most cross-sectional studies found an inverse relationship between BMI and daily behavior (as determined through physical activity). Most participants (65.2%) did not attain the recommended levels of daily behavior. This finding is consistent with previous research conducted among Spanish [15], Canadian [33], and northwest English [64] children, which demonstrated inactive lifestyles among children with overweight and obesity. Additionally, other studies also reported a more sedentary lifestyle among non-normal weight children compared to their normal weight counterparts [65,66].

Physical competence domain results showed a similar pattern as the daily behavior domains, with normal weight children attaining higher scores (16.28) and a higher number of normal weight children (31.7%) reaching the achieving and excelling levels than overweight/obese children (3.5%). Similar results were reported in previous studies that focused on the PC domain and its components, where it was found that normal weight children performed better than their overweight and obese peers on tests of muscular endurance [67], aerobic endurance [56,64,68], and fundamental motor abilities [69]. Existing non-CAPL literature also supports our findings that normal weight children show better physical competencies compared to their overweight counterparts. For example, a study conducted among Latino children in Chile and Colombia, which assessed cardiorespiratory fitness using a 20 m shuttle run test, revealed that non-normal weight children had lower cardiorespiratory fitness levels than normal weight children [70]. Other studies that also assessed physical competence in similar age groups using similar constructs of physical competence domains reported that normal weight children performed better in motor tests [71], aerobic capacity and musculoskeletal fitness [67], as well as overall fitness [72,73].

The results of the motivation and confidence domain showed that normal weight children achieved higher scores (18.37) than children with overweight and obesity, and that a significantly higher portion (30.7%) of normal weight children attained achieving and excelling levels than overweight and obese children (4.1%). In addition, the M&C domain showed that most of the normal weight and overweight/obese children were at progressing level in this domain. Lower motivation among overweight and obese children was previously reported [15,63]. Several other studies investigating the ability of healthy weight and overweight children to maintain an active lifestyle, specifically in regard to their perceived self-efficacy for physical activity, demonstrated that normal weight children tend to have fewer barriers to adopting a positive attitude toward PA than their overweight counterparts [74,75].

In the knowledge and understanding domain, children with overweight and obesity (6.43) scored higher than normal weight children (6.36), although the difference was not significant (*p* = 0.577). A higher number of normal weight children (24.1%) attained achieving and excelling levels of PL than the overweight and obese children (4.2%). Overall the majority (55.5%) of participants reached progressing level for the K&U domain. As previously discussed, the existing literature on the topic is limited in terms of determining whether healthy or unhealthy children exhibit superior performance in the domain of K&U. The available studies also reported nonsignificant differences (*p* = 0.035 [15] and *p* = 0.024 [33]) in the domains of K&U between children with normal weight and overweight and obesity.

Overall, the PL and domain scores favored normal weight children. The current study’s findings revealed a correlation that fell within the range of low to moderate. This suggests that the structure and design of the CAPL-2 assessment tool is effective when measuring the relevant variables in both normal weight and overweight and obese children and for producing valid scores for the PL and domains. The results of our study revealed a concerningly low level of PL among children in South Punjab. This finding highlights the urgent need for interventions aimed at improving the level of PL among Pakistani children. One promising approach that could be utilized in this PE endeavor is using CAPL-2 to assess and improve PL levels among school children. Additionally, a longitudinal investigation should be conducted to evaluate the long-term benefits of these interventions on the children’s overall health and well-being. This is crucial to understanding the impact of improved PL on various aspects of their development and to guide future policy-making in this area.

### Strengths and Limitations

The originality and strength of the current study lie in the fact that it was the first to utilize the well-established and validated CAPL-2 instrument to examine the PL status of children in the Pakistani population and to provide empirical evidence of the association between PL and weight status in an Asian population. Furthermore, it should be noted that the instrument has only been validated in a limited number of countries [15,42,43,44,76], and few studies have investigated the relationship between PL and weight status [15,33]. In addition, this study provides robust evidence of a positive correlation between higher levels of PL and normal weight status among Pakistani children. The findings of this study have significant implications for children’s health, by providing policymakers and health professionals with a deeper understanding of the association between PL and weight status. This research will aid in the development of evidence-based guidelines and interventions to promote physical activity engagement throughout childhood and adolescence, ultimately helping to maintain a healthy weight. In addition, this research will contribute to the existing literature, by providing a foundation for conducting future studies in other regions and populations.

Furthermore, it will be valuable for comparing PL levels with children from other countries, enabling the identification of global PL trends. This study’s strengths include an adequate sample size, which addressed the limitations of previous studies [15,43]. Another strength is the use of a combination of subjective and objective data collection methods, which accurately measured and yielded valid PL status in this population from South Punjab, Pakistan.

The current study has several limitations that warrant further investigation. One limitation is that this study only stratified the PL and domain scores according to weight status and did not consider the potential impact of other confounding factors, such as socioeconomic status, Tanner stages, pubertal status, living area, or parental influences. These variables may play a significant role in determining the causes of low PL levels, and understanding their impact could inform the development of interventions to improve PL levels among Pakistani children. Future studies should be conducted in various regions of Pakistan to determine the overall PL level in the country, as the current study was limited to the population of South Punjab. The findings of this study suggest that future research should focus on charting the PL journey by developing growth charts and establishing normative reference values for variables such as weight, height, BMI, and waist circumference using standard deviation scores (SDS) or percentile methods. A larger population sample should be utilized to further investigate the influence of potential confounding factors in future research. An in-depth analysis would be beneficial to gain a more comprehensive understanding of the relationship between PL and weight status.

Additionally, as previously suggested, longitudinal investigations are recommended, to ascertain PL levels stratified by weight status from childhood to adulthood [76]. It is important to note that the cross-sectional design used in this study precluded the establishment of a causal association between PL and weight status. Therefore, a longitudinal design should be employed in future studies, to establish a cause–effect association between the two variables.

## 5. Conclusions

In South Punjab, Pakistan, the PL and domain levels of children with normal weight were significantly higher than those of children with overweight and obesity, with the exception of the K&U domain; in this domain, children with overweight and obesity achieved slightly higher scores. The correlations between domains were weak to moderate for children with normal weight but were weak, moderate, and strong for children with overweight and obesity. The inverse correlations between BMI and domains indicate that normal weight and BMI range are strongly associated with the recommended level of PL for children. Thus, better PL and domain scores are associated with better health in children. Therefore, it is imperative that policymakers, school administration, and PE teachers prioritize the incorporation of PL into existing school PE programs for Pakistani children. This proactive approach will not only inspire children to actively participate in PA, but it will also instill healthy lifestyle habits and promote the importance of maintaining a healthy weight through increased motivation and knowledge. By prioritizing PL in school programs, we can effectively encourage Pakistani children to lead more physically active and healthy lives.

## Figures and Tables

**Figure 1 children-10-00363-f001:**
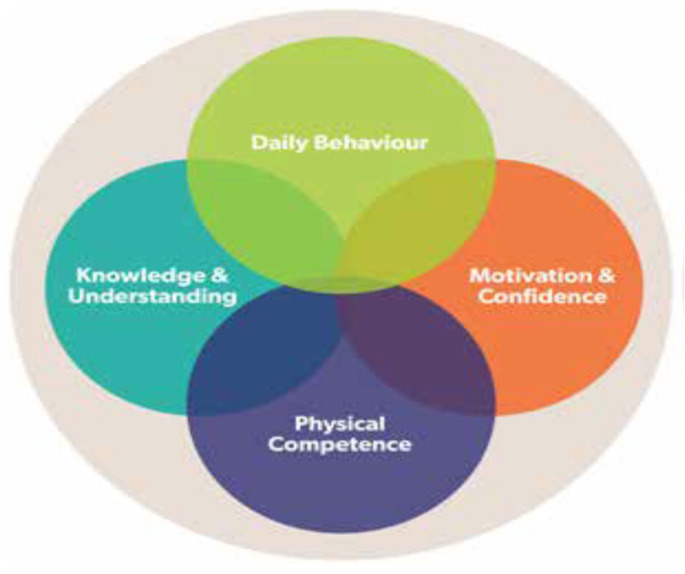
Core domains of physical literacy adapted from CAPL-2 manual HALO (2017) [31].

**Figure 2 children-10-00363-f002:**
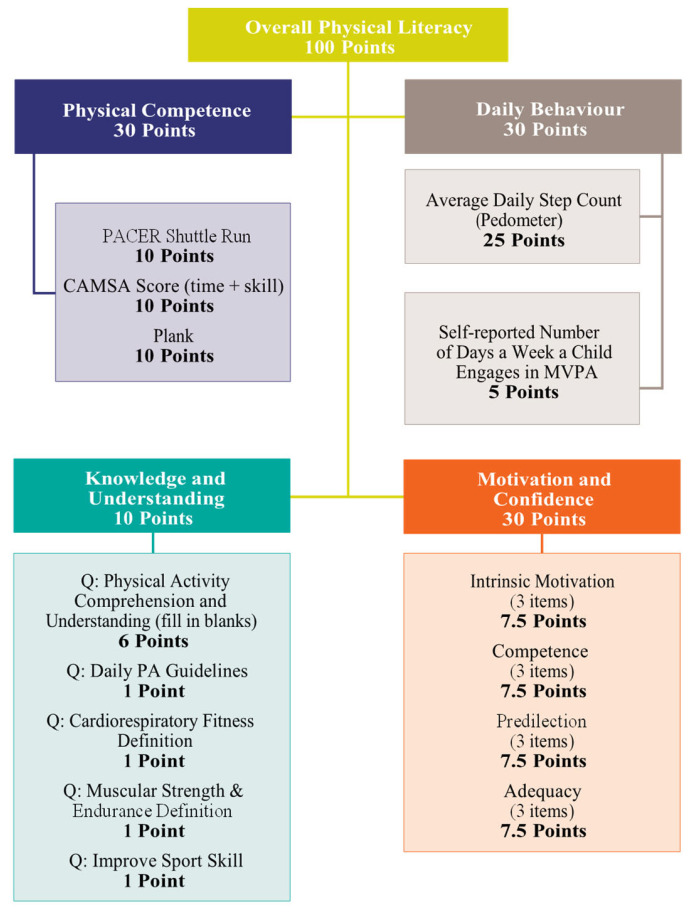
CAPL-2 domains and comprehensive scoring system adapted from CAPL-2 manual HALO (2017) [31,45,46].

**Figure 3 children-10-00363-f003:**
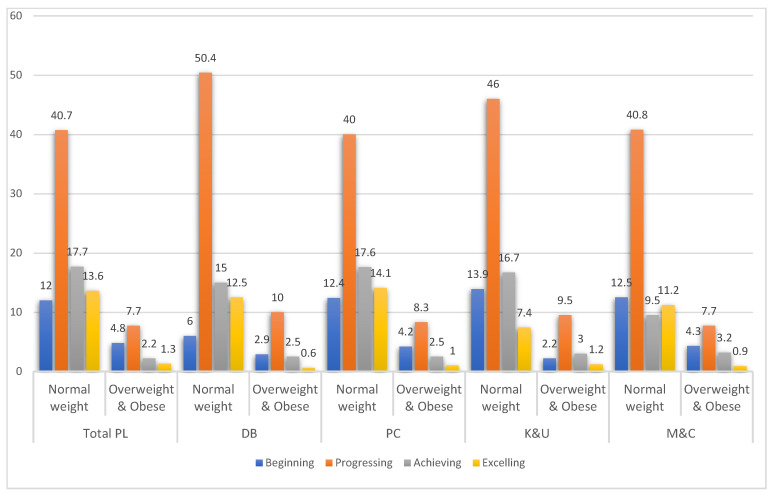
The number of children among CAPL-2 total and interpretive categories stratified by weight status.

**Table 1 children-10-00363-t001:** Descriptive characteristics of the study population.

Characteristics	Boys (%)	Girls (%)
Gender	50.2	49.8
**Age**		
8	10.3	10.0
9	10.0	10.1
10	9.5	10.1
11	9.9	9.8
12	10.5	9.8
**Grade**		
4	10.2	10.0
5	10.1	10.1
6	9.5	10.1
7	9.9	9.8
8	10.5	9.8
**City**		
Multan	16.3	16.3
Bahawalpur	17.0	16.9
D.G. Kham	16.9	16.7

Note: Data are presented as frequencies.

**Table 2 children-10-00363-t002:** Descriptive statistics for demographic and anthropometric characteristics of children.

	Boys (*n* = 675) $\bar{x}$ ± SD	Girls (*n* = 685) $\bar{x}$ ± SD	*p*-Value
Age (years)	10.00 ± 1.42	10.00 ± 1.41	0.908
Height (cm)	137.79 ± 11.42	136.74 ± 11.25	0.088
Weight (kg)	30.82 ± 8.82	30.34 ± 8.41	0.308
BMI (kg/m^2^)	16.04 ± 3.19	16.05 ± 3.14	0.920
WC (cm)	60.35 ± 9.40	58.10 ± 7.89	<0.001

Note: Data is presented as $\bar{x}$ mean, SD: standard deviation. BMI: body mass index; WC: waist circumference.

**Table 3 children-10-00363-t003:** Correlations between physical literacy domain scores in normal weight children.

	DB	PC	K&U	M&C
DB	1	-	-	-
PC	0.583 ** (0.530–0.633)	1	-	-
K&U	0.002 (−0.053–0.060)	0.001 (−0.060–0.064)	1	-
M&C	0.464 ** (0.405–0.515)	0.448 ** (0.392–0.500)	0.006 (−0.048–0.062)	1

** significant correlation at *p* < 0.01 level. *n* = 1085. 95% confidence intervals.; DB: daily behavior; PC: physical competence; K&U: knowledge and understanding; M&C: motivation and confidence.

**Table 4 children-10-00363-t004:** Correlations between physical literacy domain scores in overweight and obese children.

	DB	PC	K&U	M&C
DB	1	-	-	-
PC	0.737 ** (0.647–0.815)	1	-	-
K&U	0.089 (−0.046–0.0233)	0.104 (−0.037–0.223)	1	-
M&C	0.496 ** (0.353–0.619)	0.526 ** (0.387–0.647)	−0.023 (−0.1.52–0.108)	1

** significant correlation at *p* < 0.01 level. *n* = 206. 95% confidence intervals; DB: daily behavior; PC: physical competence; K&U: knowledge & understanding; M&C: motivation & confidence.

**Table 5 children-10-00363-t005:** Correlations between physical literacy and domain scores with body mass index (BMI).

	BMI (*n* = 1360)
Total Physical Literacy	−0.217 **
Physical Competence	−0.179 **
Knowledge & Understanding	0.006
Motivation & Confidence	−0.171 **
Daily Behavior	−0.192 **

Note. ** significant correlation at *p* < 0.01 level. Including underweight children, a total sample of 1360 was included in the correlation analysis. BMI: body mass index (kg/m^2^).

**Table 6 children-10-00363-t006:** The statistical association between weight status and anthropometric characteristics.

			Weight Status Categories				OR^+^
		Total *n* = 1291	Normal Weight *n* = 1085	Overweight and Obese *n* = 206	*p*	η_p_^2^	OR [95% CI]
Gender	Boys	648	546 (50.3%)	102 (49.5%)	0.832	-	1.66 (0.789–3.42)
	Girls	643	539 (49.7%)	104 (50.5%)			
Age (years)		10.00 ± 1.42	9.87 ± 1.41	10.66 ± 1.31	<0.001	0.042	1.36 (0.988–1.88)
Height (cm)		137.04 ± 11.30	136.28 ± 11.16	141.09 ± 11.22	<0.001	0.024	1.16 (0.693–1.95)
Weight (kg)		31.08 ± 8.53	28.86 ± 6.73	42.72 ± 7.50	<0.001	-	-
BMI (kg/m^2^)		16.34 ± 2.97	15.39 ± 2.06	21.34 ± 1.74	<0.001	-	-
WC (cm)		59.46 ± 8.74	58.33 ± 8.19	65.41 ± 9.10	<0.001	0.088	0.995 (0.952–1.04)
Wilks’ Lambda		0.450			<0.001		

Note: Data is presented as $\bar{x}$ mean, SD: standard deviation, % percentage; *p* values were obtained utilizing MONOVA (continuous variables) and a chi-squared test (categorical variable); BMI: body mass index; WC: waist circumference; OR^+^: the odd ratio was calculated using multivariate logistic regression; η_p_^2^: Partial eta squared; Effect sizes are considered small if <0.01, a medium effect <0.06, and a large effect >0.14.

**Table 7 children-10-00363-t007:** The statistical association between the weight status, PL, and domains scores.

	Weight Status Categories				OR^+^
	Normal Weight $\bar{x}$ ± SD	Overweight and Obese $\bar{x}$ ± SD	*p*	η_p_^2^	OR [95% CI]
Total Physical Literacy	52.00 ± 10.11	46.14 ± 9.36	<0.001	0.044	0.90 (0.79–1.02)
Daily Behavior	11.06 ± 3.09	9.48 ± 2.27	<0.001	0.036	0.94 (0.81–1.09)
Physical Competence	16.28 ± 3.89	14.35 ± 3.76	<0.001	0.032	1.05 (0.91–1.20)
Knowledge and Understanding	6.36 ± 1.62	6.43 ± 1.50	0.577	0.000	1.14 (0.97–1.34)
Motivation and Confidence	18.37 ± 5.56	16.05 ± 5.07	<0.001	0.024	1.07 (0.94–1.22)
Wilks’ Lambda	0.946		<0.001		

Note: Data are presented as $\bar{x}$ mean, SD: standard deviation; *p* values were obtained utilizing MONOVA; OR^+^: the odd ratio was calculated using multivariate logistic regression; η_p_^2^: Partial eta squared; Effect sizes are considered small if <0.01, a medium effect <0.06, and a large effect >0.14.

## Data Availability

Data can be requested by contacting the corresponding author on a reasonable request.

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
