# Peer review of "Assessment of the Relationship between Body Weight Status and Physical Literacy in 8 to 12 Year Old Pakistani School Children: The PAK-IPPL Cross-Sectional Study"

_children, 2023, doi:10.3390/children10020363_

Round 1
Reviewer 1 Report (Previous Reviewer 3)
I read with great interest the manuscript entitled “Assessment of the Relationship between Body Weight Status and Physical
Literacy in 8 to 12-Year-Old Pakistani School Children: A Cross-Sectional
Study”. It’s an interesting nation-based study with a large sample and much precious data. My major concern with the paper is that you didn’t segregate those with overweight from those with obesity. In addition, the ethical approval is from the school of Physical Education of Shanxi University, China; while the study was done in Pakistan.
Major comments:
Methodology:
It would have been better if the authors stratified the patients according to BMI SDS to overweight and obese children. In addition, children with underweight and those with secondary obesity and overweight should better be excluded.
The ethical approval was taken from the school of Physical Education of Shanxi University, China; although the study was done in Pakistan, please verify.
It would be nice for the authors to add the data in percentiles or SDS (e.g. weight, height, WC, and BMI percentiles) according to age and gender-matched centiles.
It would be nice for the authors to explain in the methodology the significance of the values used in the questionnaire and to interpret them in the results and discussion.
It would be nice to add data about the socioeconomic standard, educational level, dietetic history, history of diabetes, family history of obesity, diabetes, and metabolic syndrome as they may be confounding factors.
Multivariate logistic regression could explain the factors independently correlated with overweight and obesity in these children.
Discussion: The discussion is rather narrative It’s better to discuss the findings with the related studies one by one.
Minor comments:
It is preferable to use the terms children with obesity and normally weighed children than Healthy children to avoid stigmatization.
I read with great interest the manuscript entitled “Assessment of the Relationship between Body Weight Status and Physical
Literacy in 8 to 12-Year-Old Pakistani School Children: A Cross-Sectional
Study”. It’s an exciting nation-based study with a large sample and much precious data. My primary concern with the paper is that you didn’t segregate those with overweight from those with obesity. In addition, the ethical approval is from the school of Physical Education of Shanxi University, China; while the study was done in Pakistan.
Major comments:
Methodology:
It would have been better if the authors stratified the patients according to BMI SDS to overweight and obese children. In addition, children with underweight and those with secondary obesity and overweight should better be excluded.
The ethical approval was taken from the school of Physical Education of Shanxi University, China; although the study was done in Pakistan, please verify.
It would be nice for the authors to add the data in percentiles or SDS (e.g. weight, height, WC, and BMI percentiles) according to age and gender-matched centiles.
It would be nice for the authors to explain in the methodology the significance of the values used in the questionnaire and to interpret them in the results and discussion.
It would be nice to add data about the socioeconomic standard, educational level, dietetic history, history of diabetes, family history of obesity, diabetes, and metabolic syndrome as they may be confounding factors.
Multivariate logistic regression could explain the factors independently correlated with overweight and obesity in these children.
Discussion: The discussion is rather narrative It’s better to discuss the findings with the related studies one by one.
Minor comments:
It is preferable to use the terms children with obesity and normally weighed children than Healthy children to avoid stigmatization.
I read with great interest the manuscript entitled “Assessment of the Relationship between Body Weight Status and Physical
Literacy in 8 to 12-Year-Old Pakistani School Children: A Cross-Sectional
Study”. It’s an exciting nation-based study with a large sample and much precious data. My primary concern with the paper is that you didn’t segregate those with overweight from those with obesity. In addition, the ethical approval is from the school of Physical Education of Shanxi University, China; while the study was done in Pakistan.
Major comments:
Methodology:
It would have been better if the authors stratified the patients according to BMI SDS to overweight and obese children. In addition, children with underweight and those with secondary obesity and overweight should better be excluded.
The ethical approval was taken from the school of Physical Education of Shanxi University, China; although the study was done in Pakistan, please verify.
It would be nice for the authors to add the data in percentiles or SDS (e.g. weight, height, WC, and BMI percentiles) according to age and gender-matched centiles.
It would be nice for the authors to explain in the methodology the significance of the values used in the questionnaire and to interpret them in the results and discussion.
It would be nice to add data about the socioeconomic standard, educational level, dietetic history, history of diabetes, family history of obesity, diabetes, and metabolic syndrome as they may be confounding factors.
Multivariate logistic regression could explain the factors independently correlated with overweight and obesity in these children.
Discussion: The discussion is rather narrative It’s better to discuss the findings with the related studies one by one.
Minor comments:
It is preferable to use the terms children with obesity and normally weighed children than Healthy children to avoid stigmatization.
I read with great interest the manuscript entitled “Assessment of the Relationship between Body Weight Status and Physical
Literacy in 8 to 12-Year-Old Pakistani School Children: A Cross-Sectional
Study”. It’s an exciting nation-based study with a large sample and much precious data. My primary concern with the paper is that you didn’t segregate those with overweight from those with obesity. In addition, the ethical approval is from the school of Physical Education of Shanxi University, China; while the study was done in Pakistan.
Major comments:
Methodology:
It would have been better if the authors stratified the patients according to BMI SDS to overweight and obese children. In addition, children with underweight and those with secondary obesity and overweight should better be excluded.
The ethical approval was taken from the school of Physical Education of Shanxi University, China; although the study was done in Pakistan, please verify.
It would be nice for the authors to add the data in percentiles or SDS (e.g. weight, height, WC, and BMI percentiles) according to age and gender-matched centiles.
It would be nice for the authors to explain in the methodology the significance of the values used in the questionnaire and to interpret them in the results and discussion.
It would be nice to add data about the socioeconomic standard, educational level, dietetic history, history of diabetes, family history of obesity, diabetes, and metabolic syndrome as they may be confounding factors.
Multivariate logistic regression could explain the factors independently correlated with overweight and obesity in these children.
Discussion: The discussion is rather narrative It’s better to discuss the findings with the related studies one by one.
Minor comments:
It is preferable to use the terms children with obesity and normally weighed children than Healthy children to avoid stigmatization.
I read with great interest the manuscript entitled “Assessment of the Relationship between Body Weight Status and Physical
Literacy in 8 to 12-Year-Old Pakistani School Children: A Cross-Sectional
Study”. It’s an exciting nation-based study with a large sample and much precious data. My primary concern with the paper is that you didn’t segregate those with overweight from those with obesity. In addition, the ethical approval is from the school of Physical Education of Shanxi University, China; while the study was done in Pakistan.
Major comments:
Methodology:
It would have been better if the authors stratified the patients according to BMI SDS to overweight and obese children. In addition, children with underweight and those with secondary obesity and overweight should better be excluded.
The ethical approval was taken from the school of Physical Education of Shanxi University, China; although the study was done in Pakistan, please verify.
It would be nice for the authors to add the data in percentiles or SDS (e.g. weight, height, WC, and BMI percentiles) according to age and gender-matched centiles.
It would be nice for the authors to explain in the methodology the significance of the values used in the questionnaire and to interpret them in the results and discussion.
It would be nice to add data about the socioeconomic standard, educational level, dietetic history, history of diabetes, family history of obesity, diabetes, and metabolic syndrome as they may be confounding factors.
Multivariate logistic regression could explain the factors independently correlated with overweight and obesity in these children.
Discussion: The discussion is rather narrative It’s better to discuss the findings with the related studies one by one.
Minor comments:
It is preferable to use the terms children with obesity and normally weighed children than Healthy children to avoid stigmatization.
I read with great interest the manuscript entitled “Assessment of the Relationship between Body Weight Status and Physical
Literacy in 8 to 12-Year-Old Pakistani School Children: A Cross-Sectional
Study”. It’s an exciting nation-based study with a large sample and much precious data. My primary concern with the paper is that you didn’t segregate those with overweight from those with obesity. In addition, the ethical approval is from the school of Physical Education of Shanxi University, China; while the study was done in Pakistan.
Major comments:
Methodology:
It would have been better if the authors stratified the patients according to BMI SDS to overweight and obese children. In addition, children with underweight and those with secondary obesity and overweight should better be excluded.
The ethical approval was taken from the school of Physical Education of Shanxi University, China; although the study was done in Pakistan, please verify.
It would be nice for the authors to add the data in percentiles or SDS (e.g. weight, height, WC, and BMI percentiles) according to age and gender-matched centiles.
It would be nice for the authors to explain in the methodology the significance of the values used in the questionnaire and to interpret them in the results and discussion.
It would be nice to add data about the socioeconomic standard, educational level, dietetic history, history of diabetes, family history of obesity, diabetes, and metabolic syndrome as they may be confounding factors.
Multivariate logistic regression could explain the factors independently correlated with overweight and obesity in these children.
Discussion: The discussion is rather narrative It’s better to discuss the findings with the related studies one by one.
Minor comments:
It is preferable to use the terms children with obesity and normally weighed children than Healthy children to avoid stigmatization.
Author Response
Please see the attachment.

Reviewer 2 Report (New Reviewer)
I want to congratulate the authors on their meticulous job. This is a well-conducted and written study. The authors used very clear language and terminology. The concepts and ideas are nicely communicated.
I have only one minor suggestion. If I am correct, Figure 1 and Table 7 present the same data. In my opinion, the figure is more effective and could be used without table 7. However, in that case, the figure needs to be more comprehensive and accommodate more information from table 7. If updating the figure is an issue then keep only Table 7 and remove Figure 1.
Also, note that you already have Figure 1 and Figure 2 in the methods section so this should probably be Figure 3.
Good luck and take care
Author Response
Dear reviewer;
We thank you for the valuable comments and suggestions provided by you. We have considered all the points and have revised our manuscript according to them. We want to thank you for allowing us to revise our manuscript. We did our best to address the comments made by reviewers point-by-point (see below).
1- I have only one minor suggestion. If I am correct, Figure 1 and Table 7 present the same data. In my opinion, the figure is more effective and could be used without table 7. However, in that case, the figure needs to be more comprehensive and accommodate more information from table 7. If updating the figure is an issue, then keep only Table 7 and remove Figure 1.
Response: We thank the reviewer for this suggestion. The table has been removed, and we kept the figure in our manuscript.
2- Also, note that you already have Figure 1 and Figure 2 in the methods section, so this should probably be Figure 3.
Response: Thank you for pointing out this mistake; it has now been modified.
Thank you so much, Reviewer 2, for your valuable feedback and support of our study. We are truly grateful for your expertise and the time you dedicated to carefully reviewing our work.
Reviewer 3 Report (New Reviewer)
Thank you for the opportunity to review this manuscript looking at the connections between weight and physical literacy.
Below, I provide recommendations according to the different sections of your manuscript.
Introduction
The introduction is generally well written and makes the case for the study well. Having said that, I think you need to unpack physical literacy a little bit more. Two points in particular: (1) what are the fundamental movement skills associated with PL; (2) how exactly does PL support lifelong physical activity and health.
Methods
Honestly, I find the retention rate of 100% very hard to believe, especially in such a large sample and in the middle of global pandemic.
In the procedures, you say 'one school took 3 days'. Did other schools take different amounts of time? Was the intervention not standardised across all locations? If there were differences across schools, these need to be explained.
Discussion
The discussion largely re-contextualises your results and compares them to findings elsewhere. I feel that there is room for more depth and discussion of potential research/practical implications.
Further, you talk about the importance of PL and its domains in healthy lifestyle, but you only present correlations. How can we be led to believe that PL causes health, as opposed to health causing PL or its domains (e.g. motivation and confidence)?
Language/Formatting
There are numerous mistakes in the text (e.g. improper verb use, lack of punctuation) that need to be corrected. Though generally well-written, a thorough proofreading is needed.
The formatting of the sections, especially in the methods, also need to align with journal standards.
Author Response
Please see the attachment.

Reviewer 4 Report (New Reviewer)
The paper's general idea is not very novel, as it only aims to measure physical literacy in a different population than the previously measured one. However, as far as the objective is concerned, the paper is well-developed. I leave some comments below, mainly on a form that may help to improve the manuscript.
-The manuscript is very long. The authors could shorten the description of the determination of physical literacy.
-In the statistical analysis section, it is mentioned that the MANOVA test was applied. However, I need help finding the results where the MANOVA test was applied (please tell me which data were analyzed in this way).
-In the results section, some numerical values are described and then shown again in the tables. This should be corrected. I recommend that the authors remove the values from the written section and leave the tables.
-Table seven repeats information from figure 1; one should be removed.
-Improve the justification for the correlations between the physical literacy values and the test domains where this variable is obtained.
- The description of the results dominates the discussion; this section should be deepened, and comparative references to other populations should be increased.
Round 2
Reviewer 1 Report (Previous Reviewer 3)
Minor comment: It's better to replace the term non-normal weighed children with children with overweight and obesity.
Author Response
Response: We thank the reviewer for their valuable suggestion. The term non-normal weighed children have been replaced with children with overweight & obesity. It is now modified throughout the paper.
Reviewer 3 Report (New Reviewer)
Dear authors,
Thank you for the responses and comprehensive changes made. By and large, I am satisfied with the changes made.
My one major concern remaining pertains to the response rate. I understand you calculated a sample size and re-sampled to reach that size, but this is not something that means the response rate was 100%. Quite clearly, the rate was lower and you needed to re-sample. Which is fine. Please rephrase the term 'response rate' accordingly and provide additional details on how you re-sampled (e.g. how many people did you re-sample and how many responded so that you achieved your target size)
Author Response
Response: We thank the reviewer for pointing out the need for justification. As it has led to significant improvements in the sampling section. In response to the reviewer’s query, we have improved the sampling justification as suggested in the manuscript on Lines: 155-164 and in the below passage.
“A sample size of 1,360 students was randomly distributed across these 85 schools using an equal allocation method. Initially, each school was asked to provide a list of students aged 8-12 years old, from which 16 students were selected randomly. However, during the field testing phase, 11% (150 participants) of the selected students either refused to participate or completed all the tests of the CAPL-2 protocol. To maintain the sample size of 16 students per school, these 150 participants were replaced with new samples taken from the same age and schools. The final sample of 1360 included 455 participants in Multan, 455 in Bahawalpur, and 450 in Dera Ghazi Khan.”
This manuscript is a resubmission of an earlier submission. The following is a list of the peer review reports and author responses from that submission.
Round 1
Reviewer 1 Report
This study attempts to verify the relationship between the physical literacy and weight status of Pakistani children. It is thought that there is a possibility that it will develop into interesting research, but it should clearly indicate what is originality. The points to be corrected are as follows.
1. As I mentioned earlier, the originality of this research should be clearly mentioned. It should be described so that readers can understand what kind of new knowledge was obtained from this research.
2. It should be revised as a whole so that the reader can read it smoothly. For example, "PL" on line 21 should not be abbreviated. There are many similar fixes to be made, and should be resubmitted with much greater readability.
Author Response
Response to Reviewer 1 Comments
Dear reviewer;
We thank you for the valuable comments and suggestions provided by you. We have
considered all the points and have revised our manuscript according to them. We want to
thank you for offering us the opportunity to revise our manuscript. We do our best to address
the comments made by reviewers point-by-point (see below).
This study attempts to verify the relationship between the physical literacy and weight status of
Pakistani children. It is thought that there is a possibility that it will develop into interesting research,
but it should clearly indicate what is originality. The points to be corrected are as follows.
1. As I mentioned earlier, the originality of this research should be clearly mentioned. It should be
described so that readers can understand what kind of new knowledge was obtained from this
research.
Response : Thank you for this great comment. In the article text, it has been added on Lines:
515-531.
“The originality and significance of the current study relied on the fact that it is the first study
to apply the reliable and valid CPLA-2 instrument to the Pakistani population to determine
the PL status of children and the first study in an Asian population to provide empirical
evidence of the association between PL and weight status. CPLA-2 has only been validated
in five countries [15,36-38,47]; thus, only two studies have investigated the relationship
between PL level and weight status[15,27]. In addition, the recent study provided evidence
of a positive correlation between Pakistani children’s PL and healthy weight. The findings of
this study will have larger implications on children’s health by enabling policymakers and
health professionals to understand this association (PL and weight). This research will help
them develop guidelines and interventions that will encourage children to engage in PA
throughout their lives and keep their weight at a healthy level. In addition, this research will
pave the way for future studies to be conducted in other regions and populations, and it will
be valuable for comparing PL levels among children from other countries to determine global
PL trends.”
The lesson, therefore, from an educational viewpoint, is that we need to strengthen physical
education curricula by adopting physical literacy.
2. It should be revised as a whole so that the reader can read it smoothly. For example, “PL” on line
21 should not be abbreviated. There are many similar fixes to be made, and should be resubmitted
with much greater readability.
Response : We thank you for having identified these inaccuracies. Major revision has been
made throughout the paper.
Dear reviewer, please see the attachment.

Reviewer 2 Report
This seems to be a needed area of research on the topic and has good scientific processes. However, I suggest the following changes be made prior to publication. Therefore, my suggestion is minor revisions be made then a re-submission.
Title:
-Proper English for the title is: An “investigation”.
Abstract:
-Use past tense.
-Devine “PL” before abbreviating it.
Introduction:
-What do you mean by “urban obesity rates are proven to be higher than in developed countries”? Urban is in developed countries.
-Line 53: Define “That”. What is “That”?
-Lines 56-58: These seem out of place. Remove or edit these sentences or put them elsewhere in the introduction.
-Line 59: Add the abbreviation for physical literacy after it is used here.
-Lines 68-69: This does not make sense that obesity is a significant determinant of PL. You mean inverse relationship?
-Physical literacy and PL are used throughout. Once physical literacy is abbreviated, use PL the remainder of the paper.
-You make the point that “PL has long been associated with weight status”. Do you mean inversely associated?
-Second, you state “PL has long been associated with weight status”, but how is the present study different than examining the association of PL and weight status? You may want to make a slightly better case for the present investigation. Perhaps be more specific throughout this paragraph as to what previous research has found and how yours is different and contributes to the current scientific body of knowledge.
Methods:
-Make sure everything is past tense. For example, line 130, you say “employ” instead of “employed”.
-Define “WC” before abbreviating.
-Lines 177-180: Are these percentages based on the population in this paper only or based on the CDC national/international percentiles?
-Lines 186-193: Add the citation for this method. Also, did the examiner stand on the right side of the participant? They should have.
-Line 201: Please define “DB” before using it as an abbreviation.
-Provide more information about how children’s physical literacy was assessed and provide reference (probably references 39-41) for this. For example, did the children take the test on their own, on-line? Was there a practitioner there with the children to help them with the test? Etc.
-Line 299: Fix the English here – “was used to determine”.
-1,360 children participated in the study and each one complied, so there was a 100% retention rate? This seems hard to believe considering they all had to wear pedometers for 7 days and complete each of these tests. Please clarify if this is incorrect.
Results:
-All of the participants were 10 years old?
-Why are some measurements in inches and some in centimeters? Make this uniform.
Discussion:
-Well-developed and written. Clean up some grammar.
-You could add a sentence or two as to the practicality of the results. You have one sentence; “Therefore, children must keep a healthy weight in order to develop PL.” Expand on this a bit.
Other:
-It is this reviewer’s opinion to review the English throughout the paper, as there are minor grammatical errors throughout the results and discussion, especially.

Author Response
Response to Reviewer 2 Comments
Dear reviewer;
We thank you immensely for the support and the opportunity to continue developing our
paper. We appreciate the valuable comments and suggestions provided by you. We have
considered all the points and have revised our manuscript according to them. We will gladly
provide more information and clarification concerning our manuscript if needed.
This seems to be a needed area of research on the topic and has good scientific processes.
However, I suggest the following changes be made prior to publication. Therefore, my
suggestion is minor revisions be made then a re-submission.
Title:
-Proper English for the title is: An “investigation”.
Response : We thank the reviewer for this suggestion. Tittle has been modified as per all
reviewers’ suggestions.
Abstract:
-Use past tense.
-Devine “PL” before abbreviating it.
Response: Thank you for pointing out that the above suggestions have been modified and
corrected.
Introduction:
-What do you mean by “urban obesity rates are proven to be higher than in developed
countries”? Urban is in developed countries.
Response : Modified and Corrected. Lines: 46-47
-Line 53: Define “That”. What is “That”?
Response : Modified and Corrected. Line: 54
-Lines 56-58: These seem out of place. Remove or edit these sentences or put them elsewhere
in the introduction.
Response : We thank the reviewer for this suggestion. It has been removed now.
-Line 59: Add the abbreviation for physical literacy after it is used here.
Response : Corrected. Line: 59
2
-Lines 68-69: This does not make sense that obesity is a significant determinant of PL. You
mean inverse relationship?
Response : Lines: 68-71. The sentence has been corrected as; “Obesity has been recognized
as a significant factor that influences the individual PL journey, which can have far-reaching
implications for the health of the individual [20]; hence, investigating how being overweight
or obese inversely affects children’s PL may encourage overweight or obese children to
engage in a more physically active lifestyle.”
-Physical literacy and PL are used throughout. Once physical literacy is abbreviated, use PL
the remainder of the paper.
Response : Thank you for your comments. It is now modified throughout the study except
where the full form is needed to increase readability.
-You make the point that “PL has long been associated with weight status”. Do you mean
inversely associated?
Response : Modified and Corrected. Lines: 91-92.
“It has been suggested that PL has been associated with healthy weight status”
-Second, you state “PL has long been associated with weight status”, but how is the present
study different than examining the association of PL and weight status? You may want to
make a slightly better case for the present investigation. Perhaps be more specific throughout
this paragraph as to what previous research has found and how yours is different and
contributes to the current scientific body of knowledge.
Response : Thank you for your comments. It is now modified to make a better understanding.
Lines: 91-100.
As mentioned in the sentence previously, “it has been only suggested that PL has been
associated with healthy weight status and other physical, behavioral, psychological, and
social factors [21,26]; only two studies explor-ing this relationship have been conducted, and
a vast research gap exists. Since the CAPL-2 is newly developed, and for this reason, more
international studies on differ-ent populations and cultures were essential for implementing
the CPLA-2 [23] as no empirical evidence was available for PL assessment in Pakistan and
the PL relation-ship with weight status in an Asian population.” This research contributed to
evidence-based scientific literature regarding PL from Pakistan.
Methods:
-Make sure everything is past tense. For example, line 130, you say “employ” instead of
“employed”.
Response : Thank you for identifying these mistakes; we have corrected them. Line: 130
-Define “WC” before abbreviating.
Response : modified.
3
-Lines 177-180: Are these percentages based on the population in this paper only or based
on the CDC national/international percentiles?
Response: yes, these percentiles are based on the CDC international percentiles. Lines: 202-
205
-Lines 186-193: Add the citation for this method. Also, did the examiner stand on the right
side of the participant? They should have.
Response : It has been modified as suggested. Lines: 195-199.
-Line 201: Please define “DB” before using it as an abbreviation.
Response : modified.
-Provide more information about how children’s physical literacy was assessed and provide
reference (probably references 39-41) for this. For example, did the children take the test on
their own, on-line? Was there a practitioner there with the children to help them with the test?
Etc.
Response : The procedure for the assessment has been improved. Lines: 160-179
According to the manuals, the individual test procedure is described in heading 2.4.3.
Physical Literacy.
-Line 299: Fix the English here – “was used to determine”.
Response: Corrected. Line: 326
-1,360 children participated in the study and each one complied, so there was a 100%
retention rate? This seems hard to believe considering they all had to wear pedometers for 7
days and complete each of these tests. Please clarify if this is incorrect.
Response : Thank you for asking this question. Below is the clarification, and it has also been
added to the text for the readers to understand better. Lines: 142-146
All participants showed a willingness to participate and complete all tasks, which
resulted in a 100% retention rate. Along with other factors: The children were eager to learn,
perform, and competition with their peers made them excited, and these kinds of activities
were completely new for them. Further, prior to the school visit, a member of the team
contacted the PE teacher to confirm the school visit and requested that the children be present
on the day of the test, as well as notified the parents about the visit and requested them to
ensure the children’s presence on the test days. Next was the pedometer, which was also an
attraction and a new experience for children. All these play an important role in getting a
100% participant retention rate.
Lines: 174-179- Yes, we did not receive the seven-day pedometer data of 1,360 students, as
the CAPL-2 provides the solution for the missing data. The pedometer’s missing days data
were computed using the procedures described in the CAPL-2 manual [25;p34]. Further, If a
4
participant could not attend school on the assessment day, we evaluated the missing test on
the next day (since the data collection team visited each school at least four times) or
informed the date for test evaluation on the pedometer return day.
Reference 25: Healthy Active Living and Obesity Research Group (HALO). Canadian
assessment of physical literacy: Manual for test ad-ministration (2nd ed.). (2017a). Ottawa,
ON, Canada. Available online: https://www.capl-eclp.ca/wp-content/uploads/2017/10/capl-
2-manual-en.pdf (accessed on 17 January 2022).
Results:
-All of the participants were 10 years old?
Response : The study participants’ age range was 8 to 12 years old, so the mean age of the
participant was 10 years old.
-Why are some measurements in inches and some in centimeters? Make this uniform.
Response : Thank you for identifying these mistakes; we have corrected them.
Discussion:
-Well-developed and written. Clean up some grammar.
-You could add a sentence or two as to the practicality of the results. You have one sentence;
“Therefore, children must keep a healthy weight in order to develop PL.” Expand on this a
bit.
Response : Thank you for your appreciation and for having identified these inaccuracies.
Major revision has been made in the discussion part.
Lines: 559-561- Therefore, policymakers, school administration, and parents must develop
the programs by incorporating PL into already established school physical education
programs for children. This will help children become motivated to engage in physical
activity and develop healthy lifestyle habits and increase their understanding of the
significance of maintaining a healthy weight.
Other:
-It is this reviewer’s opinion to review the English throughout the paper, as there are minor
grammatical errors throughout the results and discussion, especially.
Response : Thank you for having identified these inaccuracies. Major revision has been made
throughout the paper.
➢ We thank you for these excellent comments and suggestions. We fully agreed on revisions
and updated the current version.
Dear reviewer, please see the attachment.

Reviewer 3 Report
Comments to the Authors,
I read with great interest the manuscript entitled “An Investigating of the Relationship Between Weight Status 2 and Physical Literacy of 8 to 12 Year Old Pakistani Children 3
”. This study is a big multicentric study involving 1360 children. My major concern with the paper is that the authors have not mentioned the Tanner stage and the socioeconomic status of the patients and controls and if they were matched as they may be confounding factors. Moreover, the study has many linguistic errors. Thorough English revision is needed.
Comments:
Title: The title needs thorough linguistic revision. I suggest “Assessment of the Relationship Between Weight Status 2 and Physical Literacy of 8 to 12 Year Old Pakistani Children”.
Abstract: The abbreviations should be written in details when used the first time e.g. physical literacy (PL). Please add the p values for the results. The wording of overweight and obese children as unhealthy weight is misleading as weight problems include both overweight and underweight. I suggest using the word children with overweight and obesity.
Results:
Line 309, height is described as 137.79 (Inches)I, later on in table 1 it is mentioned as cm, plz clarify.
It is better to add weight, height, BMI and waist circumference SDS.
What about the pubertal status of these children and their socioeconomic status. Adding data about this could enrich the results.
Multivariate logistic regression could be beneficial toassess the mist independent variables related to physical literacy.
Author Response
Response to Reviewer 3 Comments
Dear reviewer;
We appreciate your comments on our manuscript. All of these suggestions were helpful and
insightful, and they have a significant impact on how we might revise and improve our
manuscript. We carefully considered the suggestions and made changes that we hope will be
accepted. The paper’s key corrections and the responses to the reviewer’s comments are as
following:
I read with great interest the manuscript entitled “An Investigating of the Relationship Between Weight
Status and Physical Literacy of 8 to 12 Year Old Pakistani Children”. This study is a big multicentric
study involving 1360 children. My major concern with the paper is that the as authors have not
mentioned the Tanner stage and the socioeconomic status of the patients and controls and if they
were matched as they may be confounding factors. Moreover, the study has many linguistic errors.
Thorough English revision is needed.
Response: We thank reviewer 3 for the support of the study. We are very grateful for your
expertise and time that has clearly gone into reviewing our work.
We agree that confounding factors are highly relevant; however, our study aimed to
determine the PL status among healthy and Overweight children and further explore the
association between PL and the domains of these two groups. This study cannot cover all the
aspect and have some limitation.
Consequently, focusing on Tanner stages of children and their socioeconomic status will give
this research a new direction that is not within the scope of the present study. These aspects
have been added to future studies’ recommendations and limitations.
Proofreading has been completed, and major modification has been made.
Comments:
1- Title: The title needs thorough linguistic revision. I suggest “Assessment of the Relationship
Between Weight Status and Physical Literacy of 8 to 12 Year Old Pakistani Children”.
Response : We thank you for this suggestion. The title has been revised and modified as per
your suggestion.
Abstract:
1- The abbreviations should be written in details when used the first time e.g. physical literacy (PL).
Response : Thank you for pointing out this mistake; it has now been modified.
2- Please add the p values for the results.
Response : Thank you p values have been added on Line: 28.
3- The wording of overweight and obese children as unhealthy weight is misleading as weight
problems include both overweight and underweight. I suggest using the word children with overweight
and obesity.
Response : Thank you for this great comment. It has now been added in the text on Lines:
205-209; below is the text.
2
“For the specific purpose of classifying the PL and domain scores according to weight status,
only two categories, healthy children (normal weight) and overweight (Overweight and obese
children as one group), were used in the subsequent analyses of the PL and domain scores.
The underweight children were removed because of a small percentage (n = 69), and a similar
practice was employed in an earlier study [15,27]”.
Results:
1- Line 309, height is described as 137.79 (Inches)I, later on in table 1 it is mentioned as cm, plz
clarify.
Response: Thank you for pointing out this mistake; it has now been modified.
2- It is better to add weight, height, BMI and waist circumference SDS.
Response: thank you for this suggestion.
From the Health care point of view, calculating the SDscores is the best option, which will
help get the children’s growth idea and allow us to develop growth charts. Standard deviation
scores for height, weight, and BMI must be calculated by age and gender, but in the current
study, we are not presenting results according to each age; by doing this at this stage, our
study will go in a different direction.
Previous studies also prefer not to report SDscores while only focusing on the relationship
[27]. So for future studies in the text, this has been suggested on Lines: 539-542 as a future
recommendation.
3- What about the pubertal status of these children and their socioeconomic status. Adding data about
this could enrich the results.
Response: Thank you for highlining this point. It will enhance the usefulness of the current
study, but as this study is establishing the PL level and exploring the relationship, we could
not cover all aspects and were limited to the study objective. Confounding factors such as
SES, pubertal status, living area, and parental influences has been added to the current study
limitation. On Lines: 533-538.
4- Multivariate logistic regression could be beneficial to assess the mist independent variables related
to physical literacy.
Response: Thank you for this suggestion. We have included this in our recommendations for
the design of future studies of this nature on Lines: 542-545.
“To further investigate the influence of potential confounding factors in future research with
a larger population, it is recommended to do an in-depth analysis, such as partial correlations
and regression analysis”.
The current study only focuses on PL status and relationship among PL and domains with
weight statuses and BMI. Smiler statistical analysis has been previously considered adequate
in a study exploring the association [15].
➢ Finally, sincere thanks to reviewer 3 for your excellent comments and suggestions.
Dear reviewer, please see the attachment.

Round 2
Reviewer 1 Report
It is an improvement over the previous manuscript. However, there are still some sentences where correlation (cross-sectional study) and causality are mixed up, and we cannot judge that it can be published.
Reviewer 3 Report
The manuscript is acceptable for me in its current form.